# Polymicrobial Biofilm Organization of *Staphylococcus aureus* and *Pseudomonas aeruginosa* in a Chronic Wound Environment

**DOI:** 10.3390/ijms231810761

**Published:** 2022-09-15

**Authors:** Cassandra Pouget, Catherine Dunyach-Remy, Chloé Magnan, Alix Pantel, Albert Sotto, Jean-Philippe Lavigne

**Affiliations:** 1Bacterial Virulence and Chronic Infections, INSERM U1047, Department of Microbiology and Hospital Hygiene, CHU Nîmes, University Montpellier, CEDEX 09, 30029 Nîmes, France; 2Bacterial Virulence and Chronic Infections, INSERM U1047, Department of Infectious Diseases, CHU Nîmes, University Montpellier, CEDEX 09, 30029 Nîmes, France

**Keywords:** biofilm, BioFlux^TM^ 200, chronic wounds, gene expression, live imaging, planktonic bacteria released, *Pseudomonas aeruginosa*, *Staphylococcus aureus*

## Abstract

Biofilm on the skin surface of chronic wounds is an important step that involves difficulties in wound healing. The polymicrobial nature inside this pathogenic biofilm is key to understanding the chronicity of the lesion. Few in vitro models have been developed to study bacterial interactions inside this chronic wound. We evaluated the biofilm formation and the evolution of bacteria released from this biofilm on the two main bacteria isolated in this condition, *Staphylococcus aureus* and *Pseudomonas aeruginosa*, using a dynamic system (BioFlux™ 200) and a chronic wound-like medium (CWM) that mimics the chronic wound environment. We observed that all species constituted a faster biofilm in the CWM compared to a traditional culture medium (*p* < 0.01). The percentages of biofilm formation were significantly higher in the mixed biofilm compared to those determined for the bacterial species alone (*p* < 0.01). Biofilm organization was a non-random structure where *S. aureus* aggregates were located close to the wound surface, whereas *P. aeruginosa* was located deeper in the wound bed. Planktonic biofilm-detached bacteria showed decreased growth, overexpression of genes encoding biofilm formation, and an increase in the mature biofilm biomass formed. Our data confirmed the impact of the chronic wound environment on biofilm formation and on bacterial lifecycle inside the biofilm.

## 1. Introduction

Chronic wounds are a growing medical problem, resulting in high morbidity and mortality, costing the global healthcare system millions of dollars annually. Wounds are considered chronic when the healing process fails to proceed normally and when the anatomic function and integrity of the skin are not achieved within approximately 6 weeks [1]. Chronic wound healing is partly hampered by the presence of a polymicrobial biofilm on a complex wound exudate associating pathogenic and commensal bacteria [2,3]. Between 60 to 80% of bacteria present in these wounds are organized in biofilm [4]. Microorganisms residing in these biofilms exhibit phenotypes distinct from planktonic cells, making treatment a major challenge [5].

A major challenge is to understand the complex interactions that occur between the bacterial species inside the biofilm. Interactions between different bacterial species and the effect of the microenvironment affect bacterial behavior and virulence and thus the outcome of wound infections [5,6]. *Staphylococcus aureus* and *Pseudomonas aeruginosa* are the two main bacteria isolated in chronic wounds [7], and previous studies have evaluated their interaction [8]. For example, co-infection with these two bacteria was associated with higher inflammatory responses, increased antimicrobial tolerance, and contributed to the chronicity of the wounds [9,10,11]. Moreover, the spatial organization formed by these bacteria could influence their behavior and is a key to understanding bacterial interactions inside polymicrobial biofilm [12]. *P. aeruginosa* is localized deeper than *S. aureus* and produces virulence factors that maintain a stable and persistent inflammatory status [13].

The management of chronic wounds consists notably to eliminate the biofilm and disrupt bacterial cooperation by the use of a large debridement. Another important step is based on the control of bacterial dissemination. Indeed, studies have described the characteristics of planktonic biofilm-detached bacteria to understand the biofilm lifecycle [14,15,16,17]. Kaplan and Fine showed that *Aggregatibacter actinomycetemcomitans* biofilm colonies are capable of releasing single cells or small clusters of cells into a liquid medium and these biofilm-detached cells can attach to the surface of vessels and form new faster biofilm colonies, enabling the biofilm to spread [14]. The difference in planktonic bacteria behavior before and after the biofilm formation was also described in *P. aeruginosa* and *Klebsiella pneumoniae* [15,16,17]. Recently, we developed an in vitro medium (named Chronic Wound Medium, CWM) that closely mimics the environment encountered in chronic wounds [6]. Using this medium, we observed that the reference *S. aureus* strain Newman had reduced growth when cultivated alone, but increased growth when cocultured with *P. aeruginosa*. Moreover, *S. aureus* Newman formed a static biofilm faster and decreased its virulence in a *Caenorhabditis elegans* model after culture in CWM [6]. Taken together, our results suggested that *S. aureus* and *P. aeruginosa* could establish specific cooperation in this medium. Here, we investigated the potential of several clinical *S. aureus* and *P. aeruginosa* strains to form a polymicrobial biofilm in an open microfluidic system, the BioFlux^TM^ 200 [18] that generates a dynamic environment, using CWM. We also evaluated their spatial organization inside this mixed biofilm and the behavior of planktonic cells released from the biofilm.

## 2. Results

### 2.1. The Presence of P. aeruginosa Increased the Biofilm Formation of S. aureus in CWM

To evaluate the biofilm formation of two pairs of clinical *S. aureus* and *P. aeruginosa* strains isolated together from a diabetic foot infection, we determined the percentage of biofilm formed in the BioFlux^TM^ 200 system in conditions that reproduced those encountered in vivo by a permanent control of nutrient supply, flow, and temperature [19]. The bacteria were tested either alone (Figure 1A,B) or in a 1:1 co-culture where bacteria were added simultaneously (Figure 1C). Experiments were done in a control Brain Heart Infusion (BHI) medium and the CWM.

All strains were able to form biofilm and remain attached under shear force (Figure 1). The reference *S. aureus* Newman strain formed biofilm at each time point (Figure 1A). Interestingly, at 24-h the percentage of constituted biofilms ranged from 22.0% ± 0.2 when the strain was cultivated in BHI vs. 43.5% ± 0.4 in CWM (*p* < 0.01) (Figure 1A). This significant difference was increased at 48-h post-incubation with 43.0% ± 0.2 of biofilm in the BHI medium vs. 72.4% ± 0.5 in the CWM (*p* < 0.01) (Figure 1A). Finally, after 72-h, the percentage of formed biofilm in CMW was almost at saturation of the channel (97.3% ± 0.1) whereas it remained incomplete in BHI (68.1% ± 0.3) (*p* < 0.01) (Figure 1A). The significant difference in the biofilm formed by *S. aureus* in the CMW compared to BHI was confirmed using the clinical strains. Percentage of biofilm formed in BHI was significantly different to that formed in CWM after 24-h (24.1% ± 0.5 vs. 39.2% ± 0.6 for SAC2 and 27.3% ± 0.3 vs. 38.4% ± 0.7 for SAC4 (*p* < 0.1)), 48-h (41.1% ± 0.4 in BHI vs. 77.4% ± 0.3 in CWM for SAC2 and 43.1% ± 0.3 in BHI vs. 77.7% ± 0.2 in CWM for SAC4) and 72-h (79.5% ± 0.5 vs. 95.7% ± 0.5 for SAC2 and 81.0% ± 0.4 in BHI vs. 97.9% ± 0.8 in CWM for SAC4 (*p* < 0.01)) (Figure 1A).

The *P. aeruginosa* biofilm formation inside the BioFlux^TM^ 200 system is shown in Figure 1B. No difference in the percentage of biofilm formation of PAO1 (a reference strain) was observed after 24-h post-inoculation in the CWM (13.1% ± 0.4) compared to BHI medium (12.0% ± 0.3) (*p* = not significant (ns)) (Figure 1B). However, these percentages were significantly different at 48-h (52.4% ± 0.1 in CWM vs. 38.0% ± 0.1 in BHI; *p* < 0.1) (Figure 1B) and at 72-h (99.7% ± 0.2 vs. 70.1% ± 0.1, respectively; *p* < 0.01) (Figure 1B). The two clinical isolates, PAC2 and PAC4, showed the same trend with significantly higher percentages of biofilm formed in CMW compared to BHI after 24-h (17.3% ± 0.3 vs. 14.1% ± 0.5, respectively; *p* < 0.1), 48-h (58.2% ± 0.6 vs. 42.4% ± 0.4, respectively; *p* < 0.01) and 72-h (97.7% ± 0.2 vs. 72.1% ± 0.6, respectively; *p* < 0.01) for PAC2 (Figure 1B) and after 24-h (15.1% ± 0.2 vs. 12.1% ± 0.3, respectively; *p* < 0.1), 48-h (61.7% ± 0.2 vs. 41.7% ± 0.5, respectively, *p* < 0.01) and 72-h (98.5% ± 0.1 vs. 75.2% ± 0.5, respectively; *p* < 0.01) for PAC 4 (Figure 1B).

Interestingly, a coculture of both reference strains formed a faster and denser biofilm in CWM compared to BHI at any time (*p* < 0.01) (Figure 1C). We noted the same results for biofilm formed by both clinical pairs of SAC2/PAC2 and SAC4/PAC4 strains after 24-h (46% ± 0.4 in CWM vs. 34% ± 0.3 in BHI; *p* < 0.01), 48-h (81% ± 0.5 vs. 68% ± 0.3, respectively, *p* < 0.01) and 72-h (93% ± 0.2 vs. 75% ± 0.4, respectively; *p* < 0.01) (Figure 1C). The percentages of biofilm formation were significantly higher in the mixed biofilm compared to the bacterial species alone, whatever the medium used (*p* < 0.01).

### 2.2. P. aeruginosa and S. aureus Are Not Randomly Organized in Biofilm

In order to study the spatial distribution of these two species, we used the Live/Dead staining containing two markers, PI (dead bacteria) and Syto9, (alive bacteria) inside the BioFlux^TM^ 200 system. The staining confirmed the viability of the bacteria and distinguished cocci and bacilli inside this biofilm. The strains were cultivated in CWM for 72-h in the BioFlux^TM^ 200 before staining and fixing the cells. Three different regions on each level of the formed biofilm were examined by confocal laser scanning microscopy.

The bacteria were predominantly present as large aggregates. Using the Imaris software, we showed that most of the strains present in the biofilm were alive since the percentage of bacteria PI positive was low (<2%) (Figure 2A,B). We discriminated between *P. aeruginosa* and *S. aureus* based on shape/height and quantified each species on the deepest stack of the image and on the highest stack. This analysis showed that the *S. aureus* aggregates were located close to the wound surface, whereas *P. aeruginosa* was located deeper in the wound bed (Figure 2C).

### 2.3. Planktonic Bacteria Released from Biofilm Demonstrated Different Behavior to Initial Sessile Bacteria

To characterize the impact of biofilm formation on *S. aureus* and *P. aeruginosa* in CWM, we compared different parameters (growth, biofilm formation, and gene expression) between initial planktonic, sessile and planktonic biofilm-detached cells.

#### 2.3.1. Impact on Bacterial Growth

All bacteria, whatever their biofilm exposure, were able to grow in the CWM. However, the two clinical *S. aureus* strains presented a significantly decreased growth of the planktonic biofilm-detached (differences of 0.6 and 0.5 log, respectively) and sessile cells (1.3 and 1 log, respectively) compared to initial bacteria (*p* < 0.01) (Figure 3 and Appendix A). This effect was lesser for *S. aureus* Newman. Indeed, no statistical difference was noted between the growth curves of planktonic biofilm-detached cells and initial bacteria, whereas a significant difference was noted between sessile cells and both others (*p* < 0.01) (Appendix A).

We observed the same trend for *P. aeruginosa* strains. The clinical strains PAC2 and PAC4 had significantly reduced growth rates for planktonic biofilm-detached (differences of 0.7 and 0.4 log, respectively) (*p* < 0.01) and sessile cells (1.2 and 1 log, respectively) (*p* < 0.01) in the stationary phase compared to initial bacteria (Figure 3B and Appendix A). The PAO1 reference strain showed no difference in growth curves for the planktonic biofilm-detached cells vs. planktonic initial bacteria (*p* = ns), whereas a significantly decreased growth was noted between sessile cells and initial bacteria (difference of 0.8 log, *p* < 0.01) (Appendix A).

#### 2.3.2. Impact on Biofilm Formation

To determine the impact of bacteria lifecycle on biofilm formation in the two species cultivated in CWM, we studied two steps: the first initial step of adhesion (using the BioFilm Ring Test^®^ (BioFilm Control, St Beauzire, France)) and the last step of mature biofilm (using the mature biofilm biomass counting).

At 3-h post-incubation, *S. aureus* Newman sessile cells had an increased ability to adhere to the magnetic beads of the BioFilm Ring Test^®^ compared to planktonic biofilm-detached (*p* < 0.1) and planktonic initial bacteria (*p* < 0.01) (Table 1). The same trend was confirmed with the clinical strains. SAC2 and SAC4 planktonic biofilm-detached cells showed significantly reduced BFI compared to planktonic initial bacteria (*p* < 0.001). Interestingly, the sessile cells had a better capacity to adhere to the magnetic beads (*p* < 0.01) (Table 1). The same results were seen for *P. aeruginosa* PAO1, PAC2, and PAC4 (Table 1). The *P. aeruginosa* strains developed a biofilm more readily after having been in biofilm compared to initial cells (*p* < 0.01). Adhesion was also significantly increased for sessile cells (*p* < 0.001).

To corroborate the change of bacteria with its ability to reform a biofilm, we evaluated the mature biofilm biomass of the different strains in single culture in the CWM. Following the results obtained with the BioFilm Ring Test^®^, *S. aureus* presented a significantly higher number of bacteria constituting the biofilm for sessile bacteria compared to planktonic initial cells (Newman: logCFU/mL 8.1 ± 0.6 vs. 4.2 ± 0.5; SAC2: 8.3 ± 0.3 vs. 4.6 ± 0.4; SAC4: 8.5 ± 0.3 vs. 3.7 ± 0.3; *p* < 0.01) (Figure 4A). The planktonic cells released from the biofilm showed an intermediate behavior, with significantly higher biofilm biomass formed compared to planktonic initial bacteria (Newman: 6 ± 0.4, *p* < 0.01; SAC2: 5.7 ± 0.2, *p* < 0.01; SAC4: 6.2 ± 0.4, *p* < 0.01) but lower than sessile cells (Figure 4A). The same trend was noted for *P. aeruginosa* with sessile cells allowing the reformation of faster and denser biofilm (PAO1: logCFU/mL 10.2 ± 0.2 vs. 6.2 ± 0.2 for planktonic initial cells; PAC2: 10.3 ± 0.1 vs. 6.5 ± 0.3, respectively; PAC4: 9.7 ± 0.4 vs. 5.9 ± 0.2, respectively; *p* < 0.001) (Figure 4B).

#### 2.3.3. Impact on Expression of Genes Involved in Biofilm Formation

To confirm the effect of lifecycle on biofilm formation, we evaluated the expression of genes involved in biofilm formation: *fnbpA*, *hla*, *spaA*, and *agrA* on *S. aureus* and *rhII*, *lasI*, *pel*, and *psl* on *P. aeruginosa*.

In all *S. aureus* strains, *fnbpA* (a gene involved in adhesion and biofilm formation) and *spaA* (a gene encoding the surface protective antigen A) genes were significantly overexpressed in planktonic cells released from the biofilm compared to initial bacteria (*p* < 0.1) (Figure 5A). These genes were even more overexpressed in sessile (*p* < 0.01) (Figure 5A). Inversely, the expression of *agrA* (a negative regulator of biofilm formation) and *hla* (a gene encoding a virulence marker) were significantly decreased in biofilm-detached cells (Figure 5B). The under-expression of these two genes was also greater in sessile cells than in planktonic released ones (*p* < 0.01) (Figure 5B).

In *P. aeruginosa*, we confirmed the same trend with the over-expression of genes involved in biofilm formation among sessile bacteria (Figure 5C). Indeed, expression of *pel* and *psl* (two genes encoding extracellular polysaccharides implicated in biofilm development) was significantly increased in planktonic cells released from the biofilm (*p* < 0.1) except for PAC2, and even higher in sessile cells (*p* < 0.01) (Figure 5C). For *rhII,* a key regulator of quorum sensing in *P. aeruginosa*, expression was highly increased in sessile bacteria (*p* < 0.01) except for PAO1 where this over-expression was more attenuated (*p* < 0.1) (Figure 5D). This expression was also increased among planktonic biofilm-detached cells although this increase was more moderate than that observed for sessile bacteria (*p* < 0.1). Finally, the same results were noted for a second regulator of quorum sensing. The expression of the *lasI* gene was significantly over-expressed in planktonic cells released from biofilm (*p* < 0.1), notably in PAO1 (*p* < 0.01), but at an intermediate level compared to gene expression found in sessile cells (*p* < 0.01) (Figure 5D).

## 3. Discussion

Biofilm formation is a crucial step in the pathophysiology of chronic wounds [5] in which a large majority of bacteria are organized in biofilms [20]. It is important to determine the organization of the biofilm in this environment but also to evaluate the behavior of these bacteria after their release.

Recently, we demonstrated that microfluidics systems represent a promising complement to the current biological assays [21]. Further validation of these tools is essential to estimate their potential to mimic clinical situations. Here, we showed that the BioFlux^TM^ 200 system was well adapted to study dynamic biofilm formation in a mixed culture and the behavior of microorganisms in conditions mimicking the chronic wound environment. It is also useful to evaluate the bacterial lifecycle inside the biofilm.

After studying single-species biofilms, research gradually turned to the complexity and interactions in multispecies biofilms [22,23]. Studies have highlighted that bacteria residing in mixed biofilms were spatially organized in response to interspecies interaction [23,24]. Metabolic interactions, leading to bacterial cooperation or competition, are ubiquitous in polymicrobial biofilms and play an important role in maintaining the diversity and stability of the microbial communities [25,26,27]. Generally, in structured environments such as biofilm, the coexistence of bacterial species is favored through beneficial interactions such as co-metabolism and coordinated interaction [28,29] as demonstrated by Harrison et al. [30]. However, several recent studies have demonstrated that those interactions were not always linked to exacerbated bacterial virulence [10,31,32,33]. Competitive interactions between either pathogenic bacteria or non-pathogenic commensal microorganisms and pathogens could reduce pathogen virulence to favor the chronicity of the infection and hijack host immune defenses [33,34].

The formation of the *S. aureus* and *P. aeruginosa* mixed biofilm is particularly interesting because these bacteria are mainly co-isolated at the level of the wound bed and their interaction further complicates biofilm eradication [35]. Thus, we observed that the biofilm formed by *S. aureus* is influenced by the environment. *S. aureus* biofilm formation was significantly increased in a medium (CWM) that mimicked the environment encountered in chronic wounds (*p* < 0.01), as previously noted in static in vitro conditions [6]. Moreover, the percentages of biofilm formation of this bacterium were significantly increased when *S. aureus* was associated with *P. aeruginosa* (*p* < 0.01). This association of pathogenic species seems to favor bacterial colonization rather than exacerbation of infection. Investigation of the mechanisms of cooperation and inhibition governing the bacterial mixed biofilms of chronic wounds should also include the spatial organization of species. To study this spatial organization, we stained our clinical *S. aureus* and *P. aeruginosa* strains to evaluate viability inside polymicrobial biofilm via the live/dead method. Then, we developed a protocol to visualize by confocal microscopy the organization of the different bacterial species in the mature biofilm formed in conditions encountered in chronic wounds. Our results confirm the in vitro findings by Fazli et al. [12], demonstrating that *P. aeruginosa* was found deeper in the wound bed than *S. aureus*. This reinforces the validity of our new in vitro dynamic model. The ability of *P. aeruginosa* to migrate deeper in the wound could be explained by the role of its type IV pili and the flagellum [36,37,38]. However, *P. aeruginosa* remains close enough to the surface to capture oxygen, essential for its survival. Further studies are necessary to understand why those bacteria are not found at the same level in chronic wounds and their significance in terms of cooperation.

During biofilm formation, previous studies have shown that sessile cells acquired different physiological characteristics that changed their metabolism compared to the planktonic initial bacteria. These modifications would particularly affect the production of exopolysaccharide, bacterial growth, expression of genes regulating cell adhesion and biofilm formation, and the acquisition of resistant markers to antimicrobial agents [39]. Cell detachment [40] is a key element of biofilm development allowing the colonization of new surfaces [41,42]. However, very few studies have investigated the characteristics of the cells released from the biofilm, notably their physiology [43,44,45]. The technique developed here is particularly adapted to perform these investigations. The BioFlux^TM^ 200 system allows the collection of sessile and biofilm-detached cells and cultivation under stressful conditions. Here, we confirmed that *S. aureus* and *P. aeruginosa* modified their metabolism after biofilm formation and the modifications affected all aspects of the bacterial lifecycle after biofilm formation. We confirmed that the sessile and planktonic detached-biofilm cells were in different physiological stages compared to planktonic initial bacteria. Indeed, the cells released from biofilm represented a transition from a sessile to a planktonic phenotype. The differences observed at the end of the exponential phase of growth in sessile and biofilm-detached cells suggested that the bacteria in this state needed a period of adaptation to return to the initial potential of growth. This result confirms that the same bacteria can exhibit different profiles of virulence and are in constant adaptation to their environment [15,16,17]. Thus, studies have shown that limiting the diffusion of oxygen and nutrients in biofilms altered bacterial growth and that the biofilm-detached cells were less able to revert to the planktonic status [15,17,46]. The ability to re-adhere to surfaces and reform a biofilm also proved that sessile and biofilm-detached bacteria were able to adhere faster, as previously noted in *P. aeruginosa* [15] and in *Klebsiella pneumoniae* [17]. This potential adhesion of planktonic bacteria released from a biofilm could be linked to the stress state of the cells due to environmental conditions. To combat this stress, biofilm formation represents a protective mechanism. The modification of bacterial behavior could be explained by altered physicochemical properties of the bacteria due to activation/inhibition of different regulation virulence pathways. This modification results in an easier adhesion potential to various supports, as shown on hydrophobic surfaces [47] or an affinity for non-polar solvents [48,49]. Finally, the main question is whether the biofilm-detached cells temporarily modified their phenotypes before gradually regaining a planktonic phenotype [44,50] or if they constituted a new bacterial population [16].

## 4. Materials and Methods

### 4.1. Bacterial Strains and Culture Conditions

The bacteria and media used in this study are listed in Appendix A.

Bacteria were grown overnight in bacterial culture tubes under agitation at 200 rpm, in aerobic conditions at 37 °C in brain heart infusion (BHI, Sigma-Aldrich, Saint-Quentin-Fallavier, France) broth or in CWM (European patent application EP21305337) previously described [6].

### 4.2. Biofilm Formation of Single- and Mixed-Culture in Microfluidic Conditions

Biofilm formation in flow conditions was performed using the microfluidic system BioFlux™ 200 (Fluxion Bioscience Inc., Alameda, CA, USA). Bacteria were plated on selective agar (Mannitol salt for *S. aureus* and Cetrimide for *P. aeruginosa*). Colonies were resuspended in 3 mL of BHI or CWM and incubated at 37 °C with shaking (220 rpm) overnight. A bacterial suspension was then prepared from this overnight culture standardized to an OD_600_ of 0.1 ± 0.05 following serial 1:200 dilutions [51]. The channel was primed with 500 µL of medium alone in the inflow well with a pressure setting of 1 dyne/cm^2^ for 10 min. The remaining medium was then withdrawn. Thereafter, the microfluidic channels were inoculated by injecting the bacterial suspension from the output reservoir for 30 min at 1 dyne/cm^2^. The setup was placed on the heating plate at 37 °C. Finally, the bacterial suspension was added to the inflow well for 72-h with a pressure of 0.2 dyne/cm^2^ at 37 °C. Biofilms were obtained with bacteria cultivated either alone or in mixed culture in BHI and CWM.

### 4.3. Quantification of Biofilm Biomass and Visualization of Biofilm

The mature biofilms were evaluated using bacterial quantification of biofilm biomass. The optical density at 600 nm after 6-h of incubation of bacterial suspensions, in CWM, was adjusted to 1.00 ± 0.05, before a 1:100 dilution in CWM. Two hundred microliters of each suspension were transferred to a microplate (Falcon 96 Flat Bottom Transparent, Corning, NY, USA) in triplicate and incubated at 37 °C, 5% CO_2_ for 24-h without shaking. Negative control wells contained CWM alone. After incubation, the microplates were washed three times with 200 μL of 1X PBS. Two hundred microliters of 1X PBS was finally added into the well before biofilm disruption by sonication for 10 min at 40 kHz. Each well was then serially diluted, and the last dilution was plated on non-selective agar (LB agar). The agar plate was incubated overnight at 37 °C and CFUs were counted. The experiment was performed twice for each sample.

After 24-h, 48-h, and 72-h of incubation, biofilm formation was recorded using a fluorescence inverted microscope DM IRB (Leica Biosystems, Nanterre, France) coupled with a CoolSNAP FX camera (Roper Scientific, Lisses, France). MetaVue^TM^ software (Molecular Devices, Sunnyvale, CA, USA) was used for imaging. ImageJ^®^ was used to color black and white images, to overlay fluorescent images, to include scale bars, and calculate biofilm percentage. The 16-bit grayscale images were adjusted with the threshold function to fit the bacterial structure and were analyzed using the “Analyze Particles” function.

### 4.4. Organization of Polymicrobial Biofilm Using Confocal Microscopy

Overnight cultures in CWM were diluted to an OD_600_ of 0.1 followed by a 1:400 serial dilution into fresh medium (CWM) and combined to make an equal solution of *S. aureus* and *P. aeruginosa*. A microscopy slide was added to the microfluidic channel of the BioFlux^TM^ 200 system to remove mature biofilm formed in the channel. We let biofilm form for 72-h. Then, the microscopy slide was carefully removed and washed twice in 1X PBS before staining. Staining with PI 20 mM and Syto9 3.34 mM stock solutions in DMSO was carried out according to the BacLight^TM^ Bacterial Viability Kit manual (Invitrogen^TM^ Thermo Fisher Scientific, Waltham, MA, USA). The final concentrations of stains in the 1:1 stain mixture in PBS were 30 µM PI and 5 µM Syto9. Stain mixture was added to surfaces with biofilms (15 µL PBS-diluted stain mix directly onto surfaces and covered by coverslip). The stained samples were incubated for 15 min in the dark at room temperature before fixing cells with 3% PFA. The slide was then covered with a coverslip and imaged with an inverted Olympus Fluoview FV10i confocal laser scanning microscope fitted with a Plan-Apochromat 63×/1.40 numerical aperture oil differential interference contrast (DIC) objective set to a 1.0× digital zoom. In addition to the acquisition of DIC images, a 488-nm argon laser was used to excite any Syto9 present in the cells; a 588-nm argon laser was used to excite PI fluorescence and the emissions were collected with a 600- to 650-nm band pass filter. Images collected from biofilms were rendered with the Imaris 7.0.0 software suite (Bitplane, Saint Paul, MN, USA).

### 4.5. Determination of Growth Curves of Bacteria

Sessile, planktonic initial, and planktonic cells released from biofilm were inoculated into 3 mL of CWM and grown for 3-h with shaking. Cultures were then diluted into fresh media to obtain an initial OD_600_ above the OD_600_ of media alone (1:1–1:20 dilution rate). Two hundred microliters of bacteria were then inoculated into a 96-well flat-bottom microplate (Costar). Cultures were grown at 37 °C in an automatic microplate reader (Tecan Infinite F200), under agitation at 200 rpm. OD_600_ readings were taken every hour with continuous shaking between readings. Each experiment was performed three times.

### 4.6. Kinetics of Early Biofilm Formation

The early biofilm formation was assessed using Biofilm Ring Test^®^ (BioFilm Control, Saint Beauzire, France) as previously described [52] and according to the manufacturer’s recommendations. *P. aeruginosa* and *S. aureus* strains (in different states: sessile, planktonic initial, and planktonic biofilm-detached cells) were cultivated in CWM at 37 °C for 6-h. The bacterial suspension was standardized to an OD_600_ of 1.00 ± 0.05 and diluted at 1:250 in CWM to obtain a final concentration of 4.10^6^ CFU/mL. This bacterial suspension was complemented with 1% (*v*/*v*) magnetic beads (TON004). Two hundred microliters were then added, in triplicate, into a 96-well microplate for 3-h. The plates were incubated without shaking at 37 °C. After incubation, the microplate was placed onto a magnetic block for 1 min and scanned using a custom plate reader. The images of each well before and after magnetic attraction were analyzed with the BFCE3 software generating a Biofilm Formation Index (BFI) reflecting the adhesion strength of the strains. A high BFI value (>7) indicates high mobility of beads under magnet action, corresponding to an absence of biofilm formation, while a low value (<2) corresponds to complete immobilization of beads due to sessile cells. Each experiment was performed twice in triplicate.

### 4.7. Gene Expression of Key Regulators of Biofilm Formation

mRNA levels of *spaA*, *fnbpA*, *hla, agrA* for *S. aureus* and *lasI*, *rhII*, *pel*, *psl* for *P. aeruginosa* genes were analyzed following the method previously described by Doumith et al. [53]. Briefly, total RNA from bacterial samples was extracted with Tryzol (Invitrogen) according to the manufacturer’s instructions and samples were purified with the RNeasy^®^ Mini kit (Qiagen, Courtaboeuf, France) and treated with RNase-free DNase Set (Qiagen) 30 min at 37 °C, followed by purification. All the RNA extractions were performed in triplicate. Purity and concentration were determined using the Nanodrop^TM^ 2000 spectrophotometer (Fisher Scientific, Pittsburg, PA, USA). RT-PCR assays were performed in a LightCycler^®^480 using the one-step LightCycler^®^ RNA Master SYBR Green I kit (Roche Applied Science, Meylan, France) according to the manufacturer’s protocol. The specificity of the generated PCR products was tested by melting point analysis. Amplifications were performed in triplicate from three different RNA preparations. Cycle threshold (Ct) values of the target genes were compared with the Ct values of the housekeeping *rpoD* gene for *P. aeruginosa* and the *gyrB* gene for *S. aureus*. Those genes were chosen as endogenous references for normalizing the transcription levels of the target gene. The condition where planktonic initial bacteria were cultivated in CWM was used as control and the normalized relative expressions of the studied genes in sessile and planktonic cells released from the biofilm in CWM were determined for each strain according to the equation 2-∆∆Ct, where ∆∆Ct = (Ct_gene_—Ct_housekeeping gene_) initial planktonic bacteria CWM—(Ct_gene_—Ct_housekeeping gene_) sessile or released cells in CWM. Primers used in this study are listed in Appendix A.

### 4.8. Statistical Analysis

Statistical analyses were performed using GraphPad Prism version 9.2. Percentages of biofilm formation, spatial distribution, growth curves, BioFilm Ring test^®^ assays, and quantification of biofilm formation are presented as the mean ± standard deviation. Statistics were performed using a *t*-test. Log relative fold-change in mRNA expression evaluated by qRT-PCR is expressed as the mean fold change (planktonic initial cells representing the control standardized and the reference). Statistics were performed using one-way ANOVA followed by Dunnett’s multiple comparisons test.

## 5. Conclusions

In chronic wounds, pathological biofilm formed by *S. aureus* and *P. aeruginosa* are frequent and difficult to treat. The combination of a chronic wound medium and the BioFlux^TM^ 200 microfluidic system represents a powerful tool to study biofilm formation and to explore the evolution of the biofilm-detached bacteria that represent a transition from a sessile to a planktonic phenotype. Improving the management of chronic wounds involves enhancing knowledge of the organization of polymicrobial biofilms as well as the control of its dissemination. Developing the techniques presented in this study would improve the understanding of biofilms formed in the wound bed of chronic wounds.

## Figures and Tables

**Figure 1 ijms-23-10761-f001:**
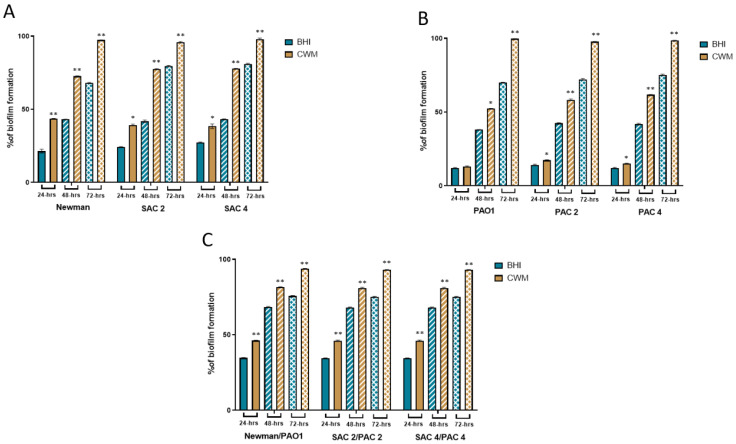
Kinetics of biofilm formation of *S. aureus* and *P. aeruginosa* strains used alone or in combination in the BioFlux^TM^ 200 system. (**A**) Percentages of biofilm formed in the microfluidic channel of the reference *S. aureus* Newman and two clinical *S. aureus* strains, SAC2 and SAC4, calculated at 24-h, 48-h, and 72-h post-incubation in BHI (blue) and CWM (yellow) media. (**B**) Percentages of biofilm formation of the reference *P. aeruginosa* PAO1 and two clinical *P. aeruginosa* strains, PAC2 and PAC4 at 24-h, 48-h, and 72-h post-incubation in BHI (blue) and CWM (yellow) media. (**C**) Percentages of biofilm formation of the coculture of *P. aeruginosa* and *S. aureus* strains at 24-h, 48-h, and 72-h post-incubation in BHI (blue) and CWM (yellow) media. Percentages of biofilm formation were calculated after three independent experiments and were determined by the software ImageJ. Results are presented as the mean ± standard deviation. Statistics were performed using a *t*-test on GraphPad Prism version 9.2 to compare the percentage of biofilm formed in BHI and CWM for each time point. * *p* < 0.1; ** *p* < 0.01.

**Figure 2 ijms-23-10761-f002:**
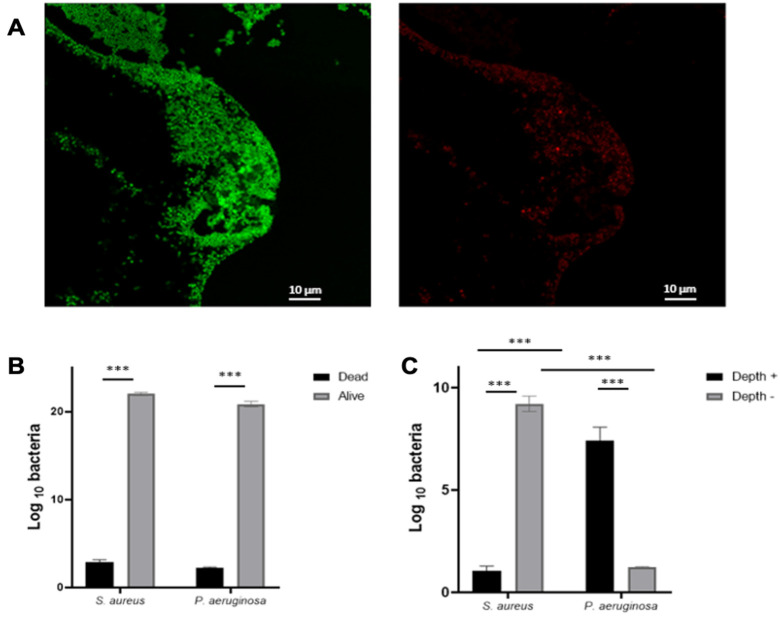
Spatial distribution of a mixed biofilm with *S. aureus* and *P. aeruginosa* inside the BioFlux^TM^ 200 system. *S. aureus* SAC2 was cocultured 1:1 with *P. aeruginosa* PAC2 cells. Cells were stained with PI/Syto9 and fixed after three days of coculture and confocal microscopy was performed to show Syto9 bacteria (alive = green; (**A**), left) and PI bacteria (dead = red; A, right) and stained bacteria were quantified (**B**). Images taken from the deepest part of the biofilm and the surface of the *P. aeruginosa*/*S. aureus* biofilm were quantified (**C**). Results are presented as the mean ± standard deviation of three different experiments. Statistics were performed using a *t*-test on GraphPad Prism version 9.2. *** *p* < 0.001.

**Figure 3 ijms-23-10761-f003:**
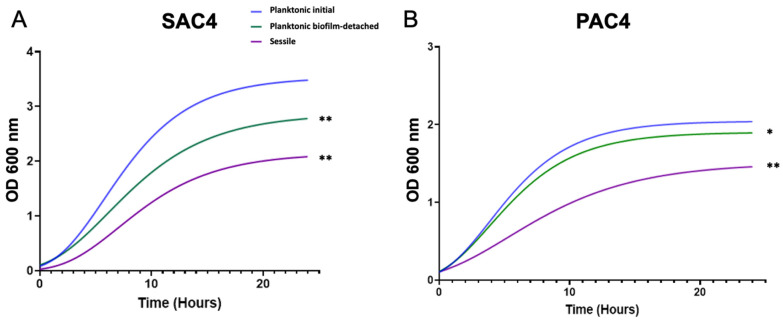
Growth curves of the clinical *S. aureus* strain SAC4 (**A**) and *P. aeruginosa* strain PAC4 (**B**) cultivated in CWM. Planktonic initial cells are represented in blue, sessile cells in purple, and planktonic biofilm-detached bacteria in green. Cultures were sampled every hour for 24-h and measurements of the OD_600_ were performed. Experiments were performed in three biological replicates. Results are presented as the mean of three different experiments. Statistics were performed using a *t*-test on GraphPad Prism version 9.2 to compare planktonic released and sessile cells compared to initial bacteria. * *p* < 0.1; ** *p* < 0.01.

**Figure 4 ijms-23-10761-f004:**
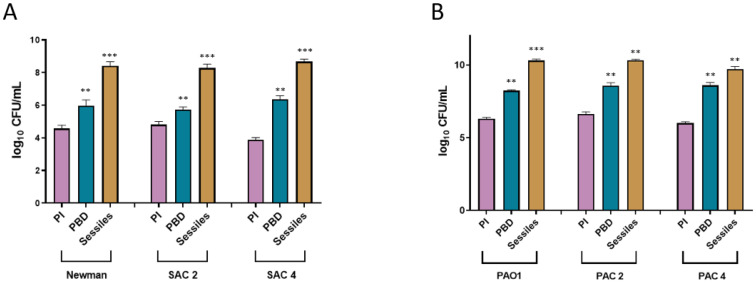
Quantification of biofilm formation of planktonic initial (pink), planktonic biofilm-detached (blue), and sessile (yellow) *S. aureus* (**A**) and *P. aeruginosa* (**B**) cultivated alone in CWM inside the open microfluidic system BioFlux^TM^ 200. The mature biofilm biomass was evaluated after 24-h incubation. Samples were tested in triplicate in two independent experiments. Results are presented as the mean ± standard deviation of three different experiments. Statistics were performed using a *t*-test on GraphPad Prism version 9.2 to compare planktonic released and sessile cells compared to planktonic initial bacteria. ** *p* < 0.01; *** *p* < 0.001.

**Figure 5 ijms-23-10761-f005:**
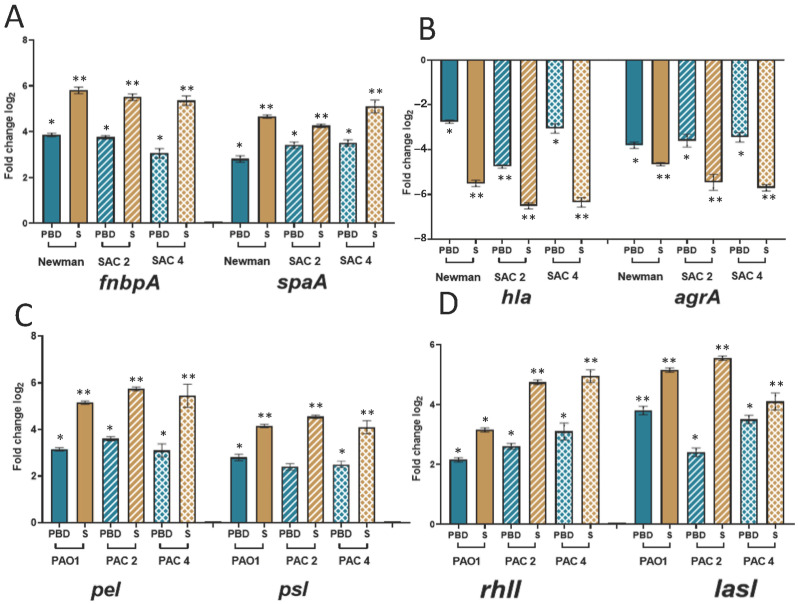
Log relative fold-change in mRNA expression by qRT-PCR of genes involved in biofilm formation for planktonic initial, planktonic biofilm-detached (PBD, blue)) and sessile (S, yellow) *S. aureus* and *P. aeruginosa* strains cultivated 72-h alone in CWM inside the open microfluidic system BioFlux^TM^ 200. Log relative fold change in expression was determined for *fnbpA*, *spaA* (**A**), *hla*, *agrA* (**B**) in *S. aureus* and *pel*, *psl* (**C**), *rhII*, *lasI*, (**D**) in *P. aeruginosa*. Reference strains are represented by solid colors, the two clinical strains are represented in squared and hatched patterns. Samples were tested in triplicate in two independent experiments. Results are expressed as the mean fold change (planktonic initial cells representing the control standardized and the reference) with error bars representing the standard deviation. Statistics were performed using one-way ANOVA followed by Dunnett’s multiple comparisons test on GraphPad Prism version 9.2. * *p* < 0.1; ** *p* < 0.01.

**Table 1 ijms-23-10761-t001:** Results of BioFilm Ring Test^®^ assays to evaluate the early biofilm formation of planktonic initial (PI), planktonic biofilm-detached (PBD) and sessile (S) *S. aureus* and *P. aeruginosa* cultivated alone in CWM inside the open microfluidic system BioFlux^TM^ 200. The Biofilm Formation Index (BFI) was evaluated after 3-h incubation. Samples were tested in triplicate in two independent experiments. Results are presented as the mean ± standard deviation of three different experiments. Statistics were performed using a *t*-test on GraphPad Prism version 9.2 to compare planktonic biofilm-detached and sessile cells compared to planktonic initial bacteria.

Strains	BFI Results			*p*	
	PI Cells	PBD Cells	S Cells	PI vs. PBD	PI vs. S
*S. aureus*					
Newman	8.7 ± 0.2	7.5 ± 0.4	6.4 ± 0.3	<0.1	<0.01
SAC2	14.9 ± 0.3	13.3 ± 0.4	10.2 ± 0.2	<0.1	<0.01
SAC4	16.7 ± 0.2	14.8 ± 0.1	10.1 ± 0.2	<0.001	<0.001
*P. aeruginosa*					
PAO1	10.4 ± 0.1	7.4 ± 0.1	4.2 ± 0.3	<0.01	<0.001
PAC2	15.5 ± 0.3	13.1 ± 0.2	7.4 ± 0.2	<0.001	<0.001
PAC4	15.0 ± 0.2	11.8 ± 0.1	7.3 ± 0.3	<0.001	<0.001

## Data Availability

Data supporting reported results can be found in the file “percentage of biofilm & evolution of bacteria inside biofilm life-cycle” contained in the computer of VBIC unit.

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
