# Peer review of "Polymicrobial Biofilm Organization of Staphylococcus aureus and Pseudomonas aeruginosa in a Chronic Wound Environment"

_ijms, 2022, doi:10.3390/ijms231810761_

Round 1

Reviewer 1 Report

Submitted manuscript entitled „Polymicrobial biofilm organization of Staphylococcus aureus 2 and Pseudomonas aeruginosa in a chronic wound environment“ focuses on characterisation of interaction of two the most often isolates of chronic wounds, Staphylococcus aureus and Pseudomonas aeruginosa. Authors used two different cultivation media, BHI and chronic-wound mimicking medium. As a model system authors used a dynamic system BioFlux™ 200.

Overall, findings of this study are interesting; however, there have been several similar studies using both types of bacteria and characterising their interaction in wound biofilm.

I have several comments and suggestion to authors in order to increase the quality of manuscript:

11.    I strongly suggest to use coloured graphs (different coloured bars in Figures). All MDPI journals are solely published online a not in printed version. Therefore, author could you coloured figures. It makes them clearer.

22.    Since chronic wound media represents important part of this manuscript, I suggest to show the exact composition of this medium and not only reference.

33.    Figure 1. It is not clear what authors mean when used expression “% of biofilm formation” and statement that “…the percentage of formed biofilm in CMW was almost complete (97.3%±0.1).“ How did you determine that biofilm was completely formed?

44.    Biofilm formation starting with bacterial adherence on surface (tissue). Howe the authors distinguish the adhered cells from the cells embedded in biofilm?

Author Response

Reviewer 1:

Overall, findings of this study are interesting; however, there have been several similar studies using both types of bacteria and characterising their interaction in wound biofilm.

We thank the reviewer for these comments and the help to improve the manuscript. If we agree that several studies have evaluated the interaction of S. aureus and P. aeruginosa notably in the biofilm formation, to our knowledge no studies have used the BioFluxTM system (and the new chronic wound medium) on clinical strains (both isolated in same chronic wounds). We also demonstrated the interest of this tool notably in the ability to evaluate the bacterial lifecycle inside the biofilm.

I have several comments and suggestion to authors in order to increase the quality of manuscript:

  1. I strongly suggest to use coloured graphs (different coloured bars in Figures). All MDPI journals are solely published online a not in printed version. Therefore, author could you coloured figures. It makes them clearer.

We modified the figures in the new version of the manuscript.

  1.   Since chronic wound media represents important part of this manuscript, I suggest to show the exact composition of this medium and not only reference.

We added the complete composition of this medium in Table S1.

  1.   Figure 1. It is not clear what authors mean when used expression “% of biofilm formation” and statement that “…the percentage of formed biofilm in CMW was almost complete (97.3%±0.1).“ How did you determine that biofilm was completely formed?

We calculated the percentage of biofilm formed in the microfluidic channel of our device (BioFluxTM 200) using ImageJ and Imaris softwares in order to calculate the percentage of the channel being covered with biofilm at a given time.

When we mentioned that the percentage of biofilm was complete, it means that the channel allowing the formation and observation of the biofilm was completely covered with biofilm. Once the channel was completely covered, the biofilm could no longer develop because it created a plug limiting the entry or the release of bacteria inside the biofilm.

We have therefore modified the text to clarify these points.

  1.   Biofilm formation starting with bacterial adherence on surface (tissue). Howe the authors distinguish the adhered cells from the cells embedded in biofilm?

We thank the reviewer for this relevant comment.

As no single technique allowed to distinguish between adhered bacteria and bacteria included in the biofilm, we studied the mechanism of biofilm formation by two approaches: i) the first one is the Biofilm Ring Test® which evaluated the bacterial adhesion to a support (magnetic beads) and not yet bacteria in biofilm. The results obtained also gave information on the speed of strains adhesion, an important parameter in the formation of the biofilm. ii) the second method is the BioFluxTM system. This method focuses on cells that have already formed biofilm and are considered sessile and/or embedded cells. Using this method, a small proportion of cells can be just adhered and are not considered in biofilm. It is important to note that this proportion of bacteria is very largely in the minority. Thus, using both techniques, we studied the bacterial behavior in biofilm distinguishing adhered cells (those that will detach but also those that will follow their life cycle in biofilm), from cells embedded inside a real biofilm.

Reviewer 2 Report

In the manuscript “Polymicrobial biofilm organization of Staphylococcus aureus and Pseudomonas aeruginosa in a chronic wound environment” of Cassandra Pouget and colleagues, the authors evaluated the biofilm formation and the evolution of bacteria released from this biofilm on the two main bacteria isolated in this condition, Staphylococcus aureus and Pseudomonas aeruginosa, using a dynamic system (BioFlux™ 200) and a chronic wound-like medium (CWM) that mimics the chronic wound environment.

The manuscript will well written and carefully conducted. This reviewer has only minor comments to improve the quality of the manuscript.

Some further information on the strains used would be helpful i.e. tying characteristics like MLST, spa-type etc for S. aureus as well as AMR data for both of them. Currently table 2 is not helpful and may be shifted into a Supplemental material section.

In Figure 5 B a different style was used why – this is confusing.

Please also make sure that for all bacterial designation the abbreviation can be used after first mentioning the full name (including tables and figure legends… please adapt this.

Check for spelling, typos and style errors (including the use of the italics style)

Author Response

Reviewer 2:

 The manuscript will well written and carefully conducted. This reviewer has only minor comments to improve the quality of the manuscript.

We thank the reviewer for this positive comment.

-Some further information on the strains used would be helpful i.e. tying characteristics like MLST, spa-type etc for S. aureus as well as AMR data for both of them

We added information in the Supplementary data (Table S1).

-. Currently table 2 is not helpful and may be shifted into a Supplemental material section.

This Table was shifted in the Supplementary data (Table S1).

-In Figure 5 B a different style was used why – this is confusing.

We corrected our typographical error with the absence of negative sign before the fold change inducing an error in the understanding of the Figure.

-Please also make sure that for all bacterial designation the abbreviation can be used after first mentioning the full name (including tables and figure legends… please adapt this. Check for spelling, typos and style errors (including the use of the italics style)

We checked all points and modified along the text.

Round 2

Reviewer 1 Report

All comments have been answered. Revised ms was improved.